# First Evidence of Activity of Enfortumab Vedotin on Brain Metastases in Urothelial Cancer Patients

**DOI:** 10.3390/ph16030375

**Published:** 2023-03-01

**Authors:** Christof Vulsteke, Laurens De Cocker, Alfonso Gómez de Liaño, Cristina Montesdeoca, Astrid De Meulenaere, Lieselot Croes, Danielle Delombaerde, Bernadett Szabados, Thomas Powles

**Affiliations:** 1Integrated Cancer Center Ghent, Department of Medical Oncology, AZ Maria Middelares, 9000 Ghent, Belgium; 2The Center for Oncological Research, University of Antwerp, 2650 Edegem, Belgium; 3Integrated Cancer Center Ghent, Department of Radiology, AZ Maria Middelares, 9000 Ghent, Belgium; 4Medical Oncology Department, CHU Insular-Materno Infantil, 35010 Las Palmas, Spain; 5Barts Cancer Institute, Queen Mary University of London, London EC1M 6BQ, UK

**Keywords:** enfortumab vedotin, brain metastases, antibody–drug conjugate

## Abstract

Enfortumab vedotin (EV), an antibody–drug conjugate directed against Nectin-4, significantly prolonged survival compared to standard chemotherapy in patients with locally advanced or metastatic urothelial carcinoma who previously received platinum-based chemotherapy and a PD-1 or PD-L1 inhibitor. The overall response rate in the phase 3 EV301 trial leading to approval was 40.6%. However, no data have been published yet regarding the effect of EV on brain metastases. Here, we present three patients from different centers with brain metastases receiving EV. A 58-year-old white male patient, who had been heavily pretreated for urothelial carcinoma with visceral metastases and a solitary clinically active brain metastasis, started on EV 1.25 mg/kg on days 1, 8, and 15 of a 28-day cycle. After three cycles, the first evaluation showed a partial remission by RECIST v1.1, with a near complete response on the brain metastasis and disappearance of neurological symptoms. The patient is currently still receiving EV. A second, 74-year-old male patient started on the same regimen, after previous progression on platinum-based chemotherapy and avelumab in maintenance. The patient achieved a complete response and received therapy for five months. Nevertheless, therapy was discontinued at the patient’s request. Shortly after, he developed new leptomeningeal metastases. Upon rechallenge with EV, there was a significant reduction in the diffuse meningeal infiltration. A third, 50-year-old white male patient also received EV after previous progression on cisplatin–gemcitabine and atezolizumab maintenance, followed by palliative whole-brain radiotherapy and two cycles of vinflunine. After three cycles of EV, there was a significant reduction in the brain metastases. The patient is currently still receiving EV. These are the first reports on the efficacy of EV in patients with urothelial carcinoma and active brain metastases.

## 1. Introduction

The standard of care for advanced urothelial carcinoma includes platinum-based chemotherapy and programmed cell death protein 1 (PD-1) or programmed cell death ligand 1 (PD-L1) inhibitors administered, respectively, as frontline, second-line, or maintenance therapy [1,2,3]. The EV-301 trial (NCT03474107) showed that enfortumab vedotin (EV) significantly prolonged survival compared to standard chemotherapy in patients with locally advanced or metastatic urothelial carcinoma who had previously received platinum-based treatment and a PD-1 or PD-L1 inhibitor [4]. EV received regulatory approval by the Food and Drug administration on 9 July 2021 for patients with locally advanced or metastatic urothelial cancer who have previously received a PD-1 or PD-L1 inhibitor and platinum chemotherapy, or who have previously received one or more prior lines of therapy if cisplatin is ineligible [5]. The European Medicines Agency approved EV on 13 April 2022 for patients with advanced or metastatic urothelial cancer and who have already received platinum-based chemotherapy and immunotherapy [6].

EV, an antibody–drug conjugate (ADC) directed against nNectin-4, is composed of a fully human monoclonal antibody specific for Nectin-4 and monomethyl auristatin E (an agent that disrupts microtubule formation). Nectin-4 is a cell-adhesion molecule that is highly expressed in urothelial carcinoma and may contribute to tumor cell growth and proliferation [7]. The targeted delivery of monomethyl auristatin E results in cell-cycle arrest and apoptosis [8].

In the registrational EV301 trial, the confirmed overall response rate was 40.6% (95% CI, 34.9 to 46.5) in the EV arm. The results of the subgroup analyses were consistent with those of the primary analysis. A complete response was observed in 4.9% of the patients (14 of 288) in the EV group and disease control was observed in 71.9% (95% CI, 66.3 to 77.0) [4]. In the EV301 trial, as in other similar trials with antibody–drug conjugates in urothelial cell cancer, patients were excluded from the trial if they had active central nervous system (CNS) metastases [9]. Subjects who had received prior treatment for CNS metastases were deemed eligible for inclusion if all of the following were true:
(1)CNS metastases have been clinically stable for at least six weeks prior to screening.(2)If steroid treatment was required for CNS metastases, the subject should be on a stable dose (≤20 mg/day) of prednisone or equivalent for at least two weeks.(3)Baseline scans show no evidence of new or enlarged brain metastases.(4)The subject does not have leptomeningeal disease [10].

However, the results of the trial did not mention subgroup analyses in patients with brain metastases, nor was it reported how many patients entered the trial with brain metastases.

In this paper, we report the first three cases on the activity of EV in urothelial cancer patients with brain metastases.

Written informed consent for the publication of this case report and any accompanying images was obtained from the three patients. A copy of the written informed consent is available for review by the Editor-in-Chief of this journal.

## 2. Case Presentations

### 2.1. Case 1

A 58-year-old white male patient underwent a nephroureterectomy and lymphadenectomy in April 2021. The pathology report showed a pT3N2 urothelial cell carcinoma of the upper tract. Cross-sectional imaging of the chest, abdomen, and pelvis showed no evidence of metastases. He was treated with four cycles of adjuvant gemcitabine and cisplatin chemotherapy. The last cycle of chemotherapy was administered on 9 July 2021. In September 2021, body computed tomography (CT) showed new nodules in both lungs and a solitary liver lesion. Subsequently, the patient received, in a platinum-refractory setting, pembrolizumab every three weeks. In January 2022, after six cycles of pembrolizumab, the patient had already progressed on CT scan. As the patient still had a good performance status, further treatment was provided. Paclitaxel was initiated in February 2022. Following two cycles of paclitaxel, progressive disease was noted by RECIST v1.1 with growth of all lung and liver metastases and the appearance of new lesions. Best supportive care was suggested; however, he remained in close follow-up as he was still without disease-related complaints. In May 2022, the patient still had a good performance status and a rechallenge was proposed with platinum-based chemotherapy. After two cycles of carboplatin–gemcitabine, the stable disease of both lung and liver metastases was noted. However, an additional CT was taken due to the recent onset of right facial numbness, which showed a nodular hypodense lesion measuring 15 mm in the axial plane localized within the junction of the left thalamus and cerebral crus of the midbrain. Following a multidisciplinary discussion, it was deemed that stereotactic radiotherapy was not possible due to the location of the lesion. Carboplatin–gemcitabine was administered for another two cycles in order to maintain systemic disease control. Re-evaluation in August 2022 showed a further enlargement of the cystic brain metastasis to 17 mm (Figure 1A). As EV became available in October 2022, the patient received EV at a dose of 1.25 mg/kg on days 1, 8, and 15 of a 28-day cycle. Prior to the second cycle, facial numbness had completely disappeared. The patient had tolerated EV quite well, with the main side effects being anorexia grade 2, maculopapular rash grade 1, peripheral neuropathy grade 1, and diarrhea grade 1. After three cycles, a partial response of the visceral lesions with a near-complete response of the thalamic brain metastasis was noted (RECIST v1.1) (Figure 1B,C, Figure 2A,B and Figure 3A,B). However, the dose had to be reduced to 1 mg/kg as of cycle four due to anorexia grade 3. The patient is currently still receiving treatment.

### 2.2. Case 2

A 74-year-old male patient presented with unexplained bowel obstruction in November 2020. A CT scan showed a primary cT3N3M1b bladder tumor, multiple pelvic and retroperitoneal lymph node metastases, as well as infiltration of the duodenum. A diagnostic sample of the bladder tumor confirmed urothelial carcinoma. The patient was started on front-line chemotherapy with gemcitabine and carboplatin and achieved a partial response at the end of chemotherapy. Subsequently, the patient received avelumab in maintenance every two weeks. In February 2022, the patient showed new left pleural metastasis and was started on EV after the failure of platinum chemotherapy and a PD-L1 inhibitor. Despite the initial response, the patient decided to withhold therapy after five months due to the side effects of EV (grade 2 fatigue and grade 1 skin toxicity). In October 2022, the patient experienced new neurological symptoms and a brain MRI confirmed a new diffuse meningeal carcinomatous infiltration of the posterior fossa and the base of the skull. The patient was rechallenged with EV and had a significant reduction in the meningeal infiltration two months later (Figure 4).

### 2.3. Case 3

A 50-year-old white male patient, who had a smoking history of 27 pack years, was diagnosed with muscle invasive bladder cancer in August 2019. Staging tests demonstrated a cT2N0M0 urothelial carcinoma. However, the patient refused neoadjuvant chemotherapy followed by cystectomy, as well as bladder preservation with trimodal treatment. After a second opinion, he underwent a repeat transurethral resection of the bladder and received intravesical instillations with Bacillus Calmette–Guérin at another center and was lost to follow-up.

In March 2021, he presented with a 2-week history of progressive dyspnea. A chest X-ray revealed a left-sided pleural effusion. In addition, the chest CT showed lung and nodal metastases as well as a solitary left ischial bone lesion, which was subsequently confirmed on a bone scan. A transbronchial biopsy was performed and urothelial carcinoma was confirmed. As his kidney function was impaired, he received six cycles of split dose cisplatin–gemcitabine in combination with atezolizumab as part of a clinical trial between May 2021 and September 2021, achieving a partial response. Atezolizumab was continued in maintenance until May 2022. The patient then presented with a mild headache, responsive to step one analgesia. A brain CT showed multiple brain metastases, while body CT showed increasing lung metastases and a new implant in his left psoas muscle. The patient was treated with palliative whole-brain radiotherapy (20 Gy in 5 fractions) and vinflunine up to 2 cycles between June 2022 and August 2022. Treatment was discontinued due to symptomatic progression (ataxia). CT scan showed no CNS response. Furthermore, an increase in lung nodules and worsening of psoas metastases was seen.

In September 2022, EV was started at a dose of 1.25 mg/kg on days 1, 8, and 15 of a 28-day cycle. After two cycles, the patient developed a grade 2 erythematous rash with associated skin sloughing. EV was withheld and a skin biopsy was taken. Topical steroids were prescribed. One week later, the rash had completely disappeared. The biopsy results ruled out Steven–Johnson Syndrome and toxic epidermal necrolysis. Thus, EV was rechallenged at a reduced dose (1 mg/kg), with no new episodes of skin toxicity. In December 2022, following three cycles of EV, the CT showed a partial response to therapy in both brain and non-CNS metastases (Figure 5, Figure 6 and Figure 7). The patient is currently still receiving treatment.

## 3. Discussion

The most common sites of distant metastases of urothelial carcinoma are the lymph nodes, liver, peritoneum, lungs, and bones. Metastases to the CNS are rare and reported in around 1–8% of patients [11]. The primary approaches to the treatment of brain metastases include surgery, stereotactic radiosurgery (SRS), or whole brain radiation. However, when surgery or SRS is not feasible, there are no data available to guide further systemic treatment. Previous research has shown that cytotoxic chemotherapies only have a limited effect on brain metastases. In addition, immune checkpoint inhibitors have also been investigated in patients with brain metastases and melanoma or lung cancer, which showed similar modest results. Thus, the data regarding the effects of anti-cancer treatment on brain metastases are very limited.

ADCs are an emerging class of anticancer drugs. They consist of tumor-targeting monoclonal antibodies linked to highly cytotoxic payloads. More than a 100 ADCs are currently being investigated in clinical trials; however, the efficacy in brain metastases is often limited due to the poor penetration of the blood–brain barrier [12]. This is also reflected in the fact that ADCs have not yet shown remarkable clinical outcomes in patients with glioblastoma multiforme [13,14,15]. A potential way to overcome these poor responses is by using ADCs with an optimal drug-to-antibody ratio, allowing for an improved payload delivery to the brain metastases. In addition, ADCs should be more homogeneously constructed as to the previous heterogeneous ADCs [12]. In heterogeneous ADCs, the drug is conjugated in random positions that can vary between the different monoclonal antibodies, whereas in homogeneous ADCs, the drug is fixed on specific sites of each monoclonal antibody molecule. For example, a post-marketing observational case series has already shown promising results of CNS activity of trastuzumab emtansine (T-DM1) in breast cancer patients [16].

Nevertheless, the registrational EV301 trial did not include patients with active CNS metastases [4]. Subjects with treated CNS metastases were able to be included given the conditions specified above. No data have been published regarding the activity of EV in patients with brain metastases up until now.

Sacituzumab govitecan (SG) is another antibody–drug conjugate that is currently being investigated in a phase 3 trial in a similar population as the EV301 trial [4,9]. SG is composed of an anti-trophoblast cell-surface antigen 2 (Trop-2) IgG1 kappa antibody coupled to SN-38, the active metabolite of irinotecan, a topoisomerase I inhibitor. This drug is already approved for triple-negative breast cancer based on the randomized phase 3 ASCENT trial. A subgroup of this trial were patients with asymptomatic brain metastases. In an exploratory analysis of these patients, SG was numerically better than the treatment physician choice (TPC) for tumor response and progression-free survival, but no overall survival. However, this benefit was clinically marginal, with an overall response rate of 3% for the SG group (1/32) vs. 0% for TPC. Moreover, no patients with active brain metastases were included [17,18]. A window of opportunity study (NCT03995706) demonstrated CNS penetrance of SG based on therapeutically relevant concentrations of its payload, SN-38, in the craniotomy tissue [19].

The three cases, who had heavily pretreated urothelial cancer, demonstrated a profound response to the brain metastases following EV. The continued exclusion of patients with active brain metastases in clinical trials with ADCs makes it difficult to assess the effect of ADCs in patients with active brain metastases. It may even reflect a missed opportunity to address a highly unmet medical need. Specific guidelines on how patients with brain metastases should be included are already available [20,21].

The cases we present here are the first reports of the activity of EV in patients with active brain metastases, thus offering a new therapeutic option in this patient population.

## Figures and Tables

**Figure 1 pharmaceuticals-16-00375-f001:**
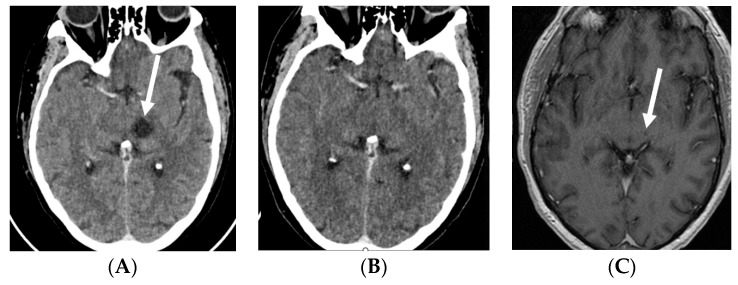
Contrast-enhanced CT scans: baseline, prior to EV treatment (**A**), and after three cycles of EV (**B**), and contrast-enhanced T1-weighted MRI after three cycles of EV (**C**). Baseline CT shows a ring-enhancing cystic/necrotic tumor in the left thalamus, whereas follow-up CT fails to show minute residual tumor. However, this is still slightly visible on the MRI scan (arrows). Abbreviations: CT, computed tomography; EV, enfortumab vedotin; MRI, magnetic resonance imaging.

**Figure 2 pharmaceuticals-16-00375-f002:**
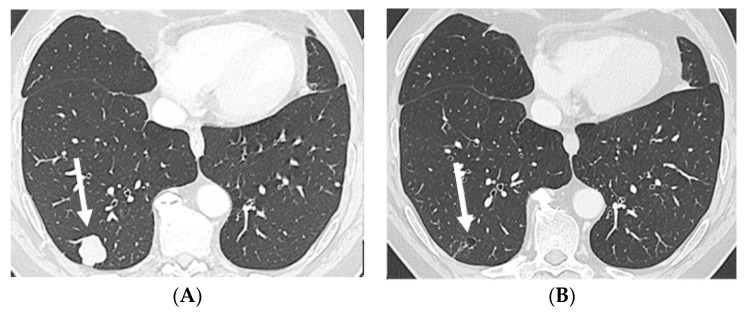
Axial CT in lung window: baseline, prior to EV treatment (**A**), and after three cycles of EV (**B**) shows a large solid metastasis in the subpleural region of the right lower lobe regressing into a small, excavated, and thin-walled air-containing cystic remnant after treatment (arrows). Abbreviations: CT, computed tomography; EV, enfortumab vedotin.

**Figure 3 pharmaceuticals-16-00375-f003:**
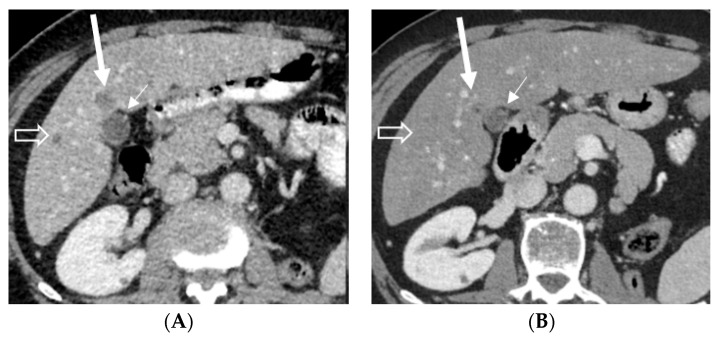
Axial contrast-enhanced abdominal CT: baseline, prior to EV treatment (**A**), and after three cycles of EV (**B**) shows almost a complete disappearance of the hypodense liver metastasis (long arrow) in front of the gallbladder (short arrow) and disappearance of a smaller metastasis (open arrow) in the right lobe after treatment. Abbreviations: CT, computed tomography.

**Figure 4 pharmaceuticals-16-00375-f004:**
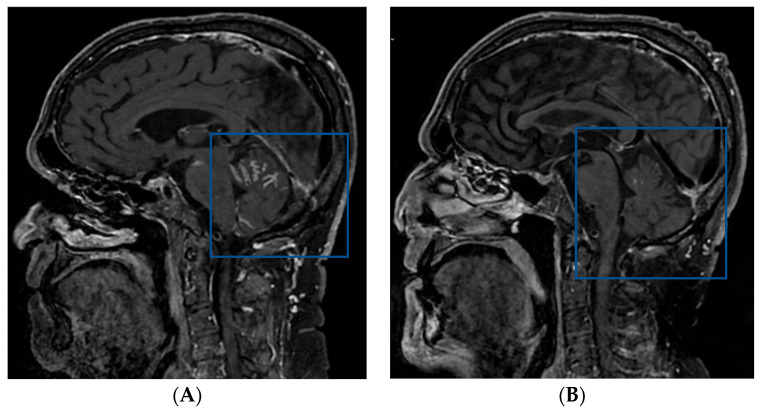
MRI scan prior to rechallenge with EV (**A**) and after two months (**B**). A significant reduction can be seen (squared boxes) of the initial meningeal carcinomatous infiltration of the posterior fossa and base of the skull. Abbreviations: EV, enfortumab vedotin; MRI, magnetic resonance imaging.

**Figure 5 pharmaceuticals-16-00375-f005:**
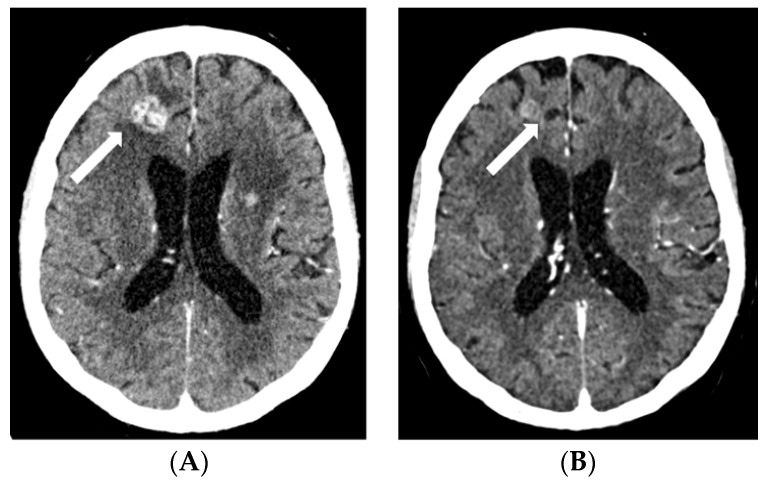
CT scan showing right frontal metastases before ((**A**), 19 mm) and after 3 cycles of EV ((**B**), 12 mm). Abbreviations: CT, computed tomography; EV, enfortumab vedotin.

**Figure 6 pharmaceuticals-16-00375-f006:**
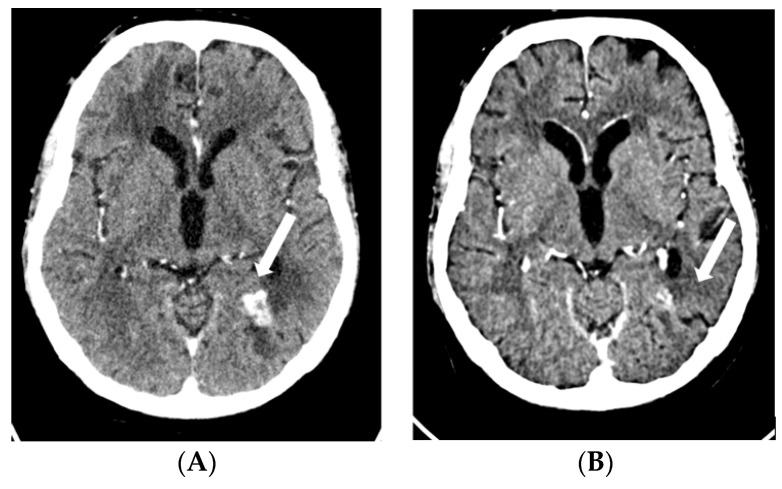
CT scan showing left parieto-occipital metastases before ((**A**), 17 mm) and after 3 cycles of EV ((**B**), 9 mm). Abbreviations: CT, computed tomography; EV, enfortumab vedotin.

**Figure 7 pharmaceuticals-16-00375-f007:**
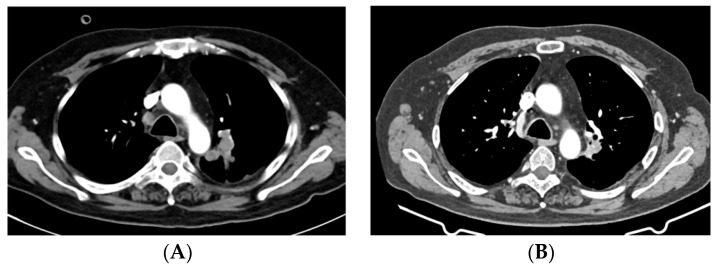
CT scan shows a left hilar lung lesion before start of EV ((**A**), 43 mm) and after 3 cycles of EV ((**B**), 23 mm). Abbreviations: CT, computed tomography; EV, enfortumab vedotin.

## Data Availability

Data is contained in the article.

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
