# Peer review of "First Evidence of Activity of Enfortumab Vedotin on Brain Metastases in Urothelial Cancer Patients"

_pharmaceuticals, 2023, doi:10.3390/ph16030375_

Round 1
Reviewer 1 Report
General comment
The manuscript entitled “First evidence of activity of enfortumab vedotin on brain metastases in urothelial patients” reports three advanced urothelial metastatic carcinomas treated with enfortumab vedotin for brain metastases. The paper is well written, readable and interesting, providing a clinical insight of the first evidence regarding the use of this drug in clinical practice. A few corrections are suggested:
INTRODUCTION
Regarding immunotherapy in bladder cancer please see: https://doi.org/10.3390/cancers14102545
The last part of the introduction could be postponed in the discussion
CASE PRESENTATION
If possible, add more data regarding the initial urothelial carcinoma, i.e. TNM and grade.
Clearly state the pathway that leads to the use of enfortumab vedotin in those patients.
DISCUSSION
Finish the manuscript with a future perspective and a summary of the importance of these findings.
Author Response
The manuscript entitled “First evidence of activity of enfortumab vedotin on brain metastases in urothelial patients” reports three advanced urothelial metastatic carcinomas treated with enfortumab vedotin for brain metastases. The paper is well written, readable and interesting, providing a clinical insight of the first evidence regarding the use of this drug in clinical practice.
We are very grateful for this nice comments.
A few corrections are suggested:
INTRODUCTION
Regarding immunotherapy in bladder cancer please see: https://doi.org/10.3390/cancers14102545
The last part of the introduction could be postponed in the discussion
CASE PRESENTATION
If possible, add more data regarding the initial urothelial carcinoma, i.e. TNM and grade.we added the lacking TNM classification
Clearly state the pathway that leads to the use of enfortumab vedotin in those patients.we added some clarifications
DISCUSSION
Finish the manuscript with a future perspective and a summary of the importance of these findings. This is a very good remark and therefore we substantially revised the discussion incorporating future perspectives and we added some data in breast cancer patients and a clear statement to allow patients wit active brain metastases into clinical trials . All can be found in the attached track changes manuscript
Reviewer 2 Report
The authors reported on cases in which EV was effective for brain metastases in patients with heavily treated urothelial carcinoma. These cases demonstrated the remarkable efficacy of EV, and the report is useful to readers regarding the treatment of an area for which there are no clinical trial data. The case presentations are well-described and easy to understand, but the discussion is poorly written. Even though this is an area where there are few data, sufficient discussion of the relationship between the blood-brain barrier and ADCs is needed, including consideration of previous papers.
Major
The discussion needs to be reorganized by referring to previous papers describing brain metastasis and ADCs (e.g., Glioblastoma or brain metastasis of breast cancer).
It would be better to include a discussion of brain metastasis in relation to previously used drugs (e.g., cisplatin, anti-PD-1 inhibitor, etc.).
Minor
Grammatical errors and typos are noticeable. Please have the manuscript thoroughly revised or professionally corrected.
Author Response
The authors reported on cases in which EV was effective for brain metastases in patients with heavily treated urothelial carcinoma. These cases demonstrated the remarkable efficacy of EV, and the report is useful to readers regarding the treatment of an area for which there are no clinical trial data. The case presentations are well-described and easy to understand, but the discussion is poorly written.
we thank the reviewer for these very kind words.
Even though this is an area where there are few data, sufficient discussion of the relationship between the blood-brain barrier and ADCs is needed, including consideration of previous papers.
We took this remark very seriously and adapted the manuscript based on this good suggestion
Major
The discussion needs to be reorganized by referring to previous papers describing brain metastasis and ADCs (e.g., Glioblastoma or brain metastasis of breast cancer).
It would be better to include a discussion of brain metastasis in relation to previously used drugs (e.g., cisplatin, anti-PD-1 inhibitor, etc.).
We incorporated all these remarks into the revised version and adpated the discussion substantially.
Minor
Grammatical errors and typos are noticeable. Please have the manuscript thoroughly revised or professionally corrected.
We revised the manuscript for grammatical errors by an native English speaking co-author
Round 2
Reviewer 2 Report
The authors have responded well to the suggestions of the reviewers and the manuscript has been improved. However, it may be excessive to state "a clear statement to allow patients wit active brain metastases into clinical trials" in the case report manuscript. This manuscript is a case report, and the author's strong assertion should be made in another manuscript or medium. The rest of the manuscript is well revised.
Author Response
The reviewer is write that we cannot state this based on case reporting We rephrased this by:
The three cases, who had heavily pretreated urothelial cancer, demonstrated a profound response to the brain metastases following EV. The continued exclusion of patients with active brain metastases in clinical trials with ADCs makes it difficult to assess the effect of ADCs in patients with active brain metastases. It may even reflect a missed opportunity to address a highly unmet medical need. Hopefully the reviewer agrees with this rephrasing. Many thanks again for the reviewers